# A Compensation Method for Nonlinearity Errors in Optical Interferometry

**DOI:** 10.3390/s23187942

**Published:** 2023-09-16

**Authors:** Yanlu Li, Emiel Dieussaert

**Affiliations:** 1Photonics Research Group, Ghent University-Imec, Technologiepark-Zwijnaarde 126, 9052 Ghent, Belgium; emiel.dieussaert@ugent.be; 2Center for Nano- and Biophotonics (NB-Photonics), Ghent University, Technologiepark-Zwijnaarde 126, 9052 Ghent, Belgium

**Keywords:** coherent detection, homodyne laser interferometry, nonlinearity errors compensation

## Abstract

Optical coherent detection is widely used for highly sensitive sensing applications, but nonlinearity issues pose challenges in accurately interpreting the system outputs. Most existing compensation methods require access to raw measurement data, making them not useful when only demodulated data are available. In this study, we propose a compensation method designed for direct application to demodulated data, effectively addressing the 1st and 2nd-order nonlinearities in both homodyne and heterodyne systems. The approach involves segmenting the distorted signal, fitting and removing baselines in each section, and averaging the resulting distortions to obtain precise distortion shapes. These shapes are then used to retrieve compensation parameters. Simulation shows that the proposed method can effectively reduce the deviation caused by the nonlinearities without using the raw data. Experimental results from a silicon-photonics-based homodyne laser Doppler vibrometry prove that this method has a similar performance as the conventional Heydemann correction method.

## 1. Background

Optical interferometry is one important technique in precise metrology, capable of discerning target displacements with nanometer or subnanometer resolution [1]. Thanks to its high accuracy and non-contact nature, optical interferometry has found widespread use in a variety of application domains such as sensing, manufacturing, medicine, and automobile industries [2]. However, the signal retrieval process of optical interferometry is prone to system deviations such as spurious reflection, beam/polarization crosstalk, flaws in optical/electrical components [3], and double-reflection from the vibrating target [4]. These factors can introduce periodic nonlinearities in the output signals. Strong nonlinearities can generate spurious sidebands, thereby complicating signal analysis and leading to data misinterpretations.

To mitigate these nonlinearity issues, various correction strategies have been proposed, typically falling into two categories. One category focuses on hardware compensation techniques, e.g., using additional compensation beams [5], introducing a rotated polarizer [6], spatially separating/misaligning beams to avoid frequency/polarization mixing [7,8], adding a tunable attenuator for laser beam vector adjustment [9], and introducing phase-encoding electronics for compensating for nonlinear errors [10]. The second category employs software-based compensation methods. One prominent numerical method is the Heydemann correction [11], which retrieves the information of the ellipse formed by the distorted in-phase-and-quadrature (IQ) signal and corrects the data based on the shape of the ellipse. This method can be used for both the homodyne [12] and heterodyne methods [13]. Other methods include a frequency-domain approach for separating the first and second harmonic nonlinearities when setting the target at a constant velocity [14], a combination of recursive weighted least-squares methods and Kalman filters [15], and a digital lock-in phase demodulator with an iterative algorithm [13]. These methods generally work well. However, most of them require access to the raw measurement data, i.e., the aforementioned IQ data. In many instances, users only have demodulated data, meaning some raw data information has been lost. As a result, the aforementioned error correction algorithms will be ineffective.

In this paper, we propose and discuss a novel numerical method to compensate for the aforementioned nonlinearities. This paper uses one typical optical interferometry—laser Doppler vibrometry (LDV) [16]—as an example for examining the methods of nonlinearity compensation. The discussed system can represent discrete optics-based, fiber-based, or photonic integrated circuit (PIC)-based interferometry.

The paper is organized as follows. Section 2 outlines the working principle of the interferometer system and the associated nonlinear problems. Section 3 introduces our proposed correction method. The detailed properties of the new method, along with their comparison to the conventional methods, particularly the Heydemann method, will be elaborated on in Section 4. The same section also discusses the effectiveness of these methods on data collected with our developed LDV system. The final part is the conclusion.

## 2. Periodic Nonlinearity in Optical Interferometry

Generally, nonlinearities are discussed separately in heterodyne and homodyne interferometry systems due to distinct structures and demodulation methods. Figure 1 illustrates typical schematic diagrams for both heterodyne and homodyne vibrometry systems and highlights the major sources of nonlinearities in each case.

Figure 1a depicts the structure of a typical heterodyne LDV based on optical fibers. The system comprises a laser diode (LD) source, a Mach–Zehnder interferometer (MZI), a photodiode (PD), and a pre-decoder. The laser source emits an optical beam with a wavelength λ. An acousto-optic modulator (AOM) is used as the splitter of the incoming beam. The 0th order diffraction of the AOM (sensing beam with a phasor *b*) is coupled to a fiber. After passing through a circulator (CIRC), the measurement light is coupled to free space and focused on the vibration target by a lens. The reflected light is collected by the same lens and re-coupled into the fiber system. Due to the Doppler effect, the reflected light carries a phase shift that is proportional to the instantaneous displacement of the target d(t) with the following relation: $\theta_{D}\left(t\right)=4\pi \cdot d\left(t\right)/\lambda$ [17]. The 1st-order diffraction of AOM (reference beam with a phasor *a*) is coupled to the same fiber system and combines with the reflected light and is detected by the PD. Thanks to the AOM, a constant optical frequency shift, $f_{s}=\omega_{s}/2\pi$, is created in the 1st-order diffraction. Therefore, the AOM acts as an optical frequency shifter (OFS) in this system. A polarization controller (PC) is used in the reference arm to ensure the polarizations of the reference and measurement signals are well aligned. The frequency shift will be converted to a carrier frequency in the photocurrent of the mixture signal. Thanks to this non-zero carrier frequency, the impact of low-frequency noise, e.g., 1/f noise, can be effectively reduced.

Given the complex amplitude reflection coefficient of the measurement light as η, the photocurrent can be expressed as [1]
(1)$$\begin{array}{ccc} i_{pd}\left(t\right) & \approx & \mu /2\cdot {\left|a\cdot e^{i\omega_{s}t}+\eta b\cdot e^{i(\theta_{D}\left(t\right)+\theta_{dc})}\right|}^{2}\\ & = & dc+\mu P_{m}\cdot \cos\left(\omega_{s}t-\theta_{D}^{\prime }\left(t\right)\right), \end{array}$$
where μ is the responsivity of the PD, $\theta_{dc}=\theta_{D}^{\prime }\left(t\right)-\theta_{D}\left(t\right)$ is the constant phase difference between the two arms due to their path length difference, $dc=\mu \left({\left|a\right|}^{2}+{\left|\eta b\right|}^{2}\right)/2$ is the dc part of the photocurrent signal, and $P_{m}=\eta \left|ab\right|$ is the amplitude of the ac part. Many methods can be used to retrieve $\theta_{D}^{\prime }\left(t\right)$; here, a commonly used method is explained [18]. In the pre-decoder, the photocurrent signals are multiplied with a pair of quadrature signals, generating two signals
(2)$$\begin{array}{ccc} I_{0}\left(t\right) & = & i_{pd}\left(t\right)\times 2\cos\omega_{s}t\\ & = & 2dc\cdot \cos\omega_{s}t+\mu P_{m}\cdot \cos\left(2\omega_{s}t-\theta_{D}^{\prime }\left(t\right)\right)+\mu P_{m}\cdot \cos\left(\theta_{D}^{\prime }\left(t\right)\right), \end{array}$$
(3)$$\begin{array}{ccc} Q_{0}\left(t\right) & = & i_{pd}\left(t\right)\times 2\sin\omega_{s}t\\ & = & 2dc\cdot \sin\omega_{s}t+\mu P_{m}\cdot \sin\left(2\omega_{s}t-\theta_{D}^{\prime }\left(t\right)\right)+\mu P_{m}\cdot \sin\left(\theta_{D}^{\prime }\left(t\right)\right). \end{array}$$

Two low-pass filters with the cut-off frequency around $f_{s}/2$ are used to remove the high-frequency components in $I_{0}\left(t\right)$ and $Q_{0}\left(t\right)$, leading to two signals:(4)$$\begin{array}{c} I\left(t\right)=\mu P_{m}\cdot \cos\left(\theta_{D}^{\prime }\left(t\right)\right), \end{array}$$(5)$$\begin{array}{c} Q\left(t\right)=\mu P_{m}\cdot \sin\left(\theta_{D}^{\prime }\left(t\right)\right). \end{array}$$

Then an arc-tangent function can be used to derive $\theta_{D}^{\prime }\left(t\right)=\arctan\left(Q(t)/I(t)\right)$. The displacement of the target Δd(t)=d(t)−d(0) can be derived as $\lambda \times (\theta_{D}^{\prime }\left(t\right)-\theta_{D}^{\prime }\left(0\right))/4\pi$.

Several issues may induce nonlinearities in such a heterodyne interferometer. The first issue is the unwanted harmonics generated by the OFS. The purpose of using an OFS is to create a constant frequency shift. AOM is the most typical OFS. It can generate several harmonics at the same time. The unwanted harmonics are usually removed by only sending the requested frequency shift to the output port of the AOM. However, a small amount of other harmonics may still find their ways to the output port, through scattering or reflecting. One can also generate a frequency shift with other phase modulators (e.g., PN junction-based phase modulator in a silicon-based photonic integrated circuit [19]) using single-sideband modulation techniques, but these methods usually suffer more from other harmonic orders. Most of these residue high-order harmonics can be removed by applying extra filters. However, some harmonics, e.g., the—1st order ($a_{-1}e^{-i\omega_{s}t}$) and 2nd order ($a_{2}e^{i2\omega_{s}t}$), cannot be completely removed by the filters and will distort the output signals. Cross-talks between the two arms may also introduce the residue harmonics to the measurement arm. Additionally, the spurious reflection (γb) in the measurement arm, usually caused by reflections in the optical interfaces, also contributes to these imperfections. Here, γ is a complex number representing the amplitude reflectivity of the total spurious reflection. Accounting for these factors and omitting small values, the multiple terms of the IQ signals can be grouped into three parts and expressed as
(6)$$\begin{array}{ccc} I(t) & = & \mu P_{m}\cdot \left[\epsilon_{1}\cdot \cos\left(\theta_{D}^{\prime }\left(t\right)+\theta_{1}\right)+\epsilon_{2}\cdot \cos\left(-\theta_{D}^{\prime }\left(t\right)+\theta_{2}\right)\right]+I_{dc}, \end{array}$$
(7)$$\begin{array}{ccc} Q(t) & = & \mu P_{m}\cdot \left[\epsilon_{1}\cdot \sin\left(\theta_{D}^{\prime }\left(t\right)+\theta_{1}\right)+\epsilon_{2}\cdot \sin\left(-\theta_{D}^{\prime }\left(t\right)+\theta_{2}\right)\right]+Q_{dc}. \end{array}$$

In these equations, the terms $\epsilon_{1}$, $\epsilon_{2}$, $\theta_{1}$, $\theta_{2}$, $I_{dc}$ and $Q_{dc}$ are introduced by the residue harmonics and cross-talks. Since only weak distortions are considered, we can assume $\epsilon_{1}>\epsilon_{2}$. This IQ Lissajous curve turns out to be an ellipse, with its center at ($I_{dc},Q_{dc}$), the long axis equal to $\mu P_{m}\left(\epsilon_{1}+\epsilon_{2}\right)$, the short axis equal to $\mu P_{m}\left(\epsilon_{1}-\epsilon_{2}\right)$, and the rotation equal to $(\theta_{1}+\theta_{2})/2$.

Figure 1b illustrates a typical homodyne system. Unlike heterodyne systems, homodyne systems do not employ an OFS. Instead, a special optical mixer is often used to obtain the quadrature signals. A typical mixer is a $90^{\circ }$ hybrid [20], which has two optical inputs and two electrical outputs. Some $90^{\circ }$ hybrids provide four electrical outputs, but they can be further combined and reduced to two signals. These two outputs can be written as
(8)$$\begin{array}{ccc} I(t) & = & \mu P_{m}\cdot \cos\left(\theta_{D}^{\prime }\left(t\right)\right), \end{array}$$
(9)$$\begin{array}{ccc} Q(t) & = & \mu P_{m}\cdot \sin\left(\theta_{D}^{\prime }\left(t\right)\right). \end{array}$$

Since there is no OFS in homodyne systems, high-order harmonics do not exist and therefore cannot be a source of nonlinearities. However, there are some different impact factors in the homodyne system, such as the imperfect phase relation in the $90^{\circ }$ hybrid and the uneven responsivities of the PDs. Taking these impact factors and the spurious reflections into account (see Figure 1b), the IQ data can be written as
(10)$$\begin{array}{ccc} I(t) & = & \mu_{1}P_{m}\cdot \cos\left(\theta_{D}^{\prime }\left(t\right)\right)+I_{dc}, \end{array}$$
(11)$$\begin{array}{ccc} Q(t) & = & \mu_{2}P_{m}\cdot \sin\left(\theta_{D}^{\prime }\left(t\right)+\theta_{r}\right)+Q_{dc}. \end{array}$$

Here, $\mu_{1}$ and $\mu_{2}$ are the responsivities of the two PDs in the first and second ports of the hybrid, and $\theta_{r}$ stands for the constant phase error in the 90${}^{\circ }$ hybrid. This IQ Lissajous curve is an ellipse, with its center at ($I_{dc},Q_{dc}$), the rotation angle $\arctan\left(\left(\Delta \pm \sqrt{\Delta^{2}+\sigma^{2}}\right)/\sigma \right)$, the semi-major and minor axis are $P_{m}\cdot {\left(\mu_{1}^{2}+\Delta \pm \sqrt{\Delta^{2}+\sigma^{2}}\right)}^{1/2}$, respectively, where $\Delta =P_{m}^{2}\cdot \left(\mu_{2}^{2}-\mu_{1}^{2}\right)/2$, and $\sigma =P_{m}^{2}\cdot \mu_{1}\mu_{2}\sin\left(\theta_{r}\right)$. When the two responsivities are the same ($\mu_{1}=\mu_{2}$), the rotation angle will be ±π/4.

These results indicate that the effects of nonlinearities bear close resemblance in both homodyne and heterodyne systems: they both distort the IQ Lissajous curves to ellipses, whereas ideally they should be circular. This is the reason why Heydemann’s correction method can be used to correct the nonlinearities in both homodyne and heterodyne systems.

To understand our proposed methods, it is important to understand how the nonlinearities distort the signals. These distortions can be categorized into two categories: the 1st-order and 2nd-order periodic nonlinearities [3]. The 1st-order nonlinearity arises when the centers of the IQ circles are not positioned at the system’s origin, while the 2nd-order nonlinearity occurs when the IQ circle becomes an ellipse rather than a circle. For example, in a fiber-based homodyne system (see Figure 1b), the 1st-order nonlinearity usually happens when the PD responsivities are not equal, while the 2nd-order nonlinearities happen when the output phase relation of the 90${}^{\circ }$ hybrid is not perfect. The schematic representation of the IQ Lissajous curves of these two types of errors can be viewed in Figure 2.

The deviated phase of the 1st-order nonlinearity can be expressed as
(12)$$\theta_{m,1st}\left(\theta_{0}\right)=\arctan\left(\frac{\sin\theta_{0}+y_{d}}{\cos\theta_{0}+x_{d}}\right),$$
where $\theta_{0}$ is the actual phase to be measured, and ($x_{d},y_{d}$) denote the shift of the IQ circle’s origin relative to the coordinate system’s origin. The phase of the 2nd-order nonlinearity can be described as
(13)$$\theta_{m,2nd}\left(\theta_{0}\right)=\arctan\left(r\cdot \tan(\theta_{0}-\theta_{d})\right)+\theta_{d},$$
where $\theta_{d}$ is the rotation of the IQ ellipse, *r* is the ratio of the semi-major axis (*a*) and the semi-minor axis (*b*) (see Figure 2).

These nonlinearities introduce an additive periodic distortion in the demodulated signal, while the periodicity happens in the demodulated phase. The 1st-order errors have a period of λ/2, while the 2nd-order errors possess a period of λ/4. Some examples of simulated distorted signals are shown in Figure 3. These plots represent extreme cases so that the periodic nonlinear errors in the demodulated phases can be easily observed. For the 1st-order deviation (Figure 3a), the deviation is caused by a change in the origin of the IQ circle. Here, the distance between the origins of the IQ circle and the coordinate system is 75% of the IQ radius. The distortions show a period of λ/2 in the displacement domain. Thanks to this phenomenon, the distortion of a signal with a larger amplitude has a higher frequency in the time domain, which can be seen in Figure 3a. For the 2nd-order nonlinear error case (Figure 3b), the deviation is only caused by the elliptical shape of the IQ circle. In this special case, the eccentricity of the ellipse ($e=\sqrt{1-{\left(b/a\right)}^{2}}$) is 0.9657 while the rotation angle is $\theta_{d}$ is π/4. The deviations have a displacement periodicity of λ/4 and their shapes are different from those of the 1st-order deviations. The vibrations in these plots have the same frequency (24 Hz) but different amplitudes (1 μm and 2 μm).

These additive deviations lead to spurious signals with a large bandwidth, which is difficult to remove with a simple filter. The frequency spectra of two sinusoidal signals ($d\left(t\right)=s_{k}\cdot \sin\left(2\pi t\times 24\;\mathrm{Hz}\right)$, $s_{1}=1\;\mu$m and $s_{2}=2\;\mu$m) with the same 2nd-order nonlinear deviation (2nd-order phase error with e=0.9657) are shown in Figure 4a. It is seen that both signals have a large number harmonics ranging from 24 Hz to kilo-hertz. A lot of useful signal in this frequency range can be influenced by the nonlinearities. Most harmonics in the spectra of $s_{2}$ are higher than those of $s_{1}$, but this is not the case for all harmonics. The frequency spectrum of the neck skin displacement induced by carotid pulses is measured by an on-chip homodyne LDV [21] and plotted in Figure 4b (red spectrum). The measurement is made on the neck of one of the authors. The measured signal is the displacement of the neck skin with a duration of 1 s. A moving-average filter is firstly applied to the measured signal to remove the existing nonlinear errors and speckle noises. Note that the moving-average filter not only removes the nonlinearity, but also removes the useful data with higher frequency. After this procedure, a 2nd-order phase error with e=0.9657 is purposely added to the pulse signal numerically. It can be seen that the spectra of the signal with the added nonlinearities (blue curve in Figure 4b) is considerably increased at the higher-frequency part. This indicates that the nonlinearity distortion in the spectrum may cause significant errors in signal analysis.

## 3. Proposed Methods

In this section, we will describe the proposed methods that can be used to compensate for the nonlinear effects directly on the demodulated data. We will introduce two methods, but the main discussion will be focus on the second one.

### 3.1. Inverse Function Method

Based on the periodic feature of the demodulated data, it is possible to retrieve the strength of the nonlinearity for certain datasets. This approach requires the data to have a strictly monotonic increase or decrease throughout its entire section. Once this criterion is met, the data can undergo an inverse mapping from y=f(x) to $x=f^{-1}\left(y\right)$. Applying a Fast-Fourier Transform (FFT) to the $f^{-1}\left(y\right)$ allows us to obtain the strength of the periodic nonlinearities. Since the deviation is not sinusoidal in the inverse function, it will have a number of sidebands in the FFT spectrum.

Signals with several different 1st-order deviations are shown in Figure 5a. Here, we employed a signal with an amplitude of 10 μm, which ensures enough periods in the rising slope (here, we assume the optical wavelength is 1550 nm). Given the monotonic nature of these signals, we can extract their inverse functions (see Figure 5b). The spectra of these inverse functions reveal the periodic feature at multiples of 2/λ in Figure 5c. With a higher deviation ($y_{d}=0.75$), the spectra exhibit stronger and more fringes. Similarly, signals with different 2nd-order deviations (eccentricities *e* = 0.5, 0.7 and 0.95), their inverse functions and the spectra of their inverse functions are shown in Figure 6a, Figure 6b, and Figure 6c, respectively. The periodic features are located at multiples of 4/λ. Simulation shows that a higher eccentricity corresponds to a stronger peak at 4/λ.

Since the frequency at 2/λ is primarily for the 1st-order error, one can use this peak as an indicator of the 1st-order error’s strength. To realize the compensation for this error, one can scan the two parameters ($x_{ds},y_{ds}$) to the measured phase $\theta_{m,1st}\left(t\right)$ based on the following phase compensation algorithm
(14)$$\theta_{scan}\left(t\right)=\arctan\left(\frac{\sin\theta_{m,1st}\left(t\right)-y_{ds}}{\cos\theta_{m,1st}\left(t\right)-x_{ds}}\right).$$

As an example, the strengths of the 2/λ peak for a signal with a 1st-order distortion ($d_{x}=0$ and $d_{y}=0.25$) under the compensation scan are shown in Figure 7. It is seen that $x_{ds}=0$ and $y_{ds}=0.25$ correspond to the lowest 2/λ peak, which are exactly the predetermined values. However, the scan demands substantial computational resources. To improve this speed, a more efficient algorithm to identify the global minimum might be beneficial.

Similarly, the 2nd-order error can also be compensated by conducting the following scan
(15)$$\theta_{scan}\left(t\right)=\arctan\left(\frac{1}{r_{s}}\cdot \tan\left(\theta_{m,2nd}\left(t\right)-\theta_{s}\right)+\theta_{s}\right),$$
where $r_{s}$ and $\theta_{s}$ are the two scan parameters. For the 2nd-order compensation, the figure of merit (FOM) should be the fringe strength of 4/λ. Since the 1st-order errors can also generate a harmonic at 4/λ, it would be better to compensate for the 1st-order error before the 2nd-order when both nonlinear errors exist.

However, this method requires several strict prerequisites. (1) The signals should be strictly monotonic and span more than several periods of wavelength. For a general signal, it requires another complex procedure to find a section that meets such criteria. (2) When strong noise exists, the signal’s monotonicity is further disrupted. Therefore, the noise should be kept low. (3) Scanning parameters is a relatively time-intensive operation. As discussed, a faster algorithm is to be developed. In practical applications, the use of the inverse-function method is not very feasible.

### 3.2. Piece-Wise-Fitting Method

To make it easier to handle more general data, we propose a second method with the following steps:

(1) Segmentation: Segment the entire displacement signal into displacement intervals of λ/2. There are two types of data segments. Some data start at 0 and end at λ/2, or start at λ/2 and end at 0, which are called “complete sections”. Other data do not reach either 0 or λ/2, which are called “incomplete sections”. The incomplete sections will be discarded in the following steps. In this step, we usually convert the displacement signal to the corresponding phase changes $\theta_{m}^{0}\left(t\right)$. After segmentation, the phase of each section can be expressed as $\theta_{mk}^{0}\left(t\right)$, where *k* is the index of the section.

(2) Baseline fitting: For each section, the target’s movement is assumed to be rapid enough that it can be approximated by either a linear function, i.e., $\theta_{bl}\left(t\right)=at+b$, or a quadratic function, i.e., $\theta_{bl}\left(t\right)=at^{2}+bt+c$. The baseline function $\theta_{bl,k}\left(t\right)$ for each complete section can be determined using numerical fitting functions (e.g., Polyfit in Matlab). Note that only strictly monotonic baseline functions will be considered in the subsequent steps.

(3) Deviation Averaging: calculate the deviation of the phase as:(16)$$devi\theta_{k}^{0}\left(t_{k}\right)=\theta_{m,k}^{0}\left(t_{k}\right)-\theta_{bl,k}\left(t_{k}\right).$$

This allows for the separation of the measured signal into the baseline and deviation signals. Thanks to the simple shape of the baseline, the nonlinear distortions can only be present in the deviation parts. Given the strict monotonicity of $\theta_{bl,k}\left(t_{k}\right)$, its inverse function $t_{k}\left(\theta_{bl,k}\right)$ exists. Therefore, the deviation can be written as a function of the baseline $\theta_{bl,k}$ phase as:(17)$$devi\theta_{k}\left(\theta_{bl,k}\right)=devi\theta_{k}^{0}\left(t_{k}\left(\theta_{bl,k}\right)\right)=\theta_{m,k}^{0}\left(t_{k}\left(\theta_{bl,k}\right)\right)-\theta_{bl,k}=\theta_{m,k}\left(\theta_{bl,k}\right)-\theta_{bl,k}.$$

Assume the baseline $\theta_{bl,k}\left(t_{k}\right)$ is the undistorted signal and $devi\theta_{k}^{0}\left(t_{k}\right)$ is the nonlinear distortion, the function $\theta_{m,k}\left(\theta_{bl,k}\right)$ will be the same as Equation (Equation 12) for the 1st-order nonlinearity and Equation (Equation 13) for the 2nd-order nonlinearity. However, this is not the case most of the time. Only some special cases, e.g., when the target is moving at a constant speed, can make this happen. However, if we can average over a large number of sections, the averaged relation will approach the theoretical relations of the nonlinearities. To facilitate this averaging, we interpolate $devi\theta_{k}\left(\theta_{bl,k}\right)$ to $devi\theta_{k}\left(\theta_{s}\right)$, where $\theta_{s}$ are an array of predefined phases ranging evenly from 0 to 2π. After interpolation, samples from all complete sections have the same independent variables $\theta_{s}$ and the averaging can be realized by calculating the following value $\bar{\theta_{m,k}\left(\theta_{s}\right)}=\sum_{k=1}^{N}\theta_{m,k}\left(\theta_{s}\right)/N$. In this case, we can retrieve an averaged deviation based on the following relation $\bar{devi\theta_{k}\left(\theta_{s}\right)}=\bar{\theta_{m,k}\left(\theta_{s}\right)}-\theta_{s}$.

(4) Deviation Fitting: Based on our assumption that the averaged deviation functions converge to the theoretical nonlinearity errors, the key deviation parameters can be retrieved by fitting the averaged deviation to the theoretical relations. For the 1st-order nonlinearity, the averaged deviation will be fitted to
(18)$$\bar{devi\theta_{k}\left(\theta_{s}\right)}\approx \arctan\left(\frac{\sin\theta_{s}+y_{d}}{\cos\theta_{s}+x_{d}}\right)-\theta_{s},$$
and the best values of $x_{d}$, $y_{d}$ will be estimated as $x_{fd}$, $y_{fd}$, respectively. For the 2nd-order nonlinearity, the averaged deviation will be fitted to
(19)$$\bar{devi\theta_{k}\left(\theta_{s}\right)}\approx \arctan\left(r\cdot \tan\left(\theta_{s}-\theta_{d}\right)+\theta_{d}\right)-\theta_{s}.$$The best values of *r*, $\theta_{d}$ will be obtained as $r_{f}$ and $\theta_{fd}$, respectively. Several methods can be used for deviation fitting. This paper demonstrates the method using Matlab’s in-built fit function.

(5) Compensation: compensate the distorted signals with the retrieved parameters ($x_{fd}$, $y_{fd}$ or $r_{f}$, $\theta_{fd}$). For the 1st-order nonlinearity, the compensated phase is calculated as
(20)$$\theta_{compensated}\left(t\right)=\arctan\left(\frac{\sin\left(\theta_{m}^{0}\left(t\right)\right)-y_{fd}}{\cos\left(\theta_{m}^{0}\left(t\right)\right)-x_{fd}}\right).$$

For the 2nd-order nonlinearity, the compensated phase is
(21)$$\theta_{compensated}\left(t\right)=\arctan\left(\frac{1}{r_{f}}\cdot \tan\left(\theta_{m}^{0}\left(t\right)-\theta_{fd}\right)+\theta_{fd}\right),$$

In this step, we can calculate the FOM defined by the root-mean-square (RMS) of the signal deviation from the actual signal $\theta_{0}\left(t\right)$, denoted as $RMS(\theta_{c}\left(t\right)-\theta_{0}\left(t\right))$, where $\theta_{c}\left(t\right)$ is the compensated values. In this paper, the RMS values are expressed with a unit of half-wavelength, allowing these results to be extended to sensors with other sensing wavelengths.

Figure 8 illustrates an example of the steps involved in compensating for 1st-order nonlinearity. In this example, a sinusoidal signal with an amplitude of 6 μm and a frequency of 24 Hz is used as the original signal. This signal is then distorted by shifting the IQ circle to $x_{d}=-0.2$ and $y_{d}=0.14$, resulting in an RMS deviation of 0.0274×λ/2. Figure 8a and Figure 8b depict the decomposition process. The entire signal in Figure 8a is segmented into seven sections. The first six sections are complete, while the final one is incomplete and, hence, discarded. Figure 8c shows the linear baseline fitting of one section, with Figure 8d displaying the obtained deviations for all complete sections. The deviations are averaged and fitted to Equation (Equation 18), as shown in Figure 8e. With the parameters of the fitted function ($x_{fd}=-0.223$ and $y_{fd}=0.132$), the distorted signal can be corrected, as illustrated in Figure 8f. After post-compensation, the RMS deviation reduces to 0.0049×λ/2. Additional compensation on the corrected signal using the same algorithm can further reduce the RMS deviation to 0.0037×λ/2, which is a seven-fold reduction.

This method can also compensate for 2nd-order nonlinearity, with the retrieved parameters being $r_{f}$ and $\theta_{fd}$. An example of this compensation is shown in Figure 9. The original signal used is the same as that in the 1st-order compensation. The deviation is due to an ellipse alteration with r=1.135 and $\theta_{d}=45^{\circ }$, leading to an RMS deviation of 0.007×λ/2. The retrieved fitting parameters are $r_{f}=1.134$ and $\theta_{fd}=45.0^{\circ }$, reducing the RMS deviation to 0.00028×λ/2—a 25-fold reduction. In this case, since the first compensation is already very effective, additional compensation steps will not further improve the results. Note that a quadratic fitting is used for baseline fitting in this case, instead of a linear fitting. A detailed discussion of the differences between linear and quadratic baseline fitting will follow in the next section.

In cases where both types of deviations are present, they are not additive. Instead, the total deviation should be expressed as
(22)$$\theta_{m,total}\left(\theta_{0}\right)=\theta_{m,1st}\left(\theta_{m,2nd}\left(\theta_{0}\right)\right),$$
or
(23)$$\theta_{m,total}\left(\theta_{0}\right)=\theta_{m,2nd}\left(\theta_{m,1st}\left(\theta_{0}\right)\right).$$

Because of this relationship, compensation for a general signal can be achieved by first applying the 1st-order (2nd-order) nonlinearity compensation to the distorted data, followed by the 2nd-order (1st-order) compensation. Note that the two orders may correspond to different deviation parameters. The effectiveness of the compensation can be different for different compensation orders, which will be discussed in the next section.

This piecewise fitting method has an advantage over the inverse-function method in that it does not require a large monotonic region of the signal. As such, it is applicable to a wider range of data. Due to the challenges in implementing the inverse-function compensation methods, the subsequent discussion will focus solely on the performance of the piecewise fitting method.

## 4. Discussion

We will present a comparison of compensations for signals with varying frequencies and amplitudes. The 1st-order deviations were introduced by incorporating $x_{d}=-0.2$ and $y_{d}=0.14$ into the signal with variable amplitudes and frequencies. The RMS deviation results after the 1st-order compensation are illustrated in Figure 10, where linear baseline fitting is applied in Figure 10a, and quadratic fitting is used in Figure 10b.

From Figure 10, we can see that the compensation works well only when the vibration amplitude (half of the peak-to-peak amplitude) is larger than 3λ/4. We also identified spikes at vibration amplitudes equating to integer multiples of λ/2. Apart from these spikes, linear fitting provided marginally superior RMS deviation values compared to the qudratic fitting. These spikes are connected to instances where the vibration signal is sinusoidal. For a more random signal, such periodic features do not appear, as evidenced by a signal with two frequency components shown in Figure 11. In this case, the spikes shown in Figure 10a disappear. It is also found that the linear baseline fitting is slightly better than the quadratic baseline fitting in the sense of better RMS deviations for higher vibration amplitudes (see Figure 11b).

For the 2nd-order nonlinearity, a similar study is conducted using sinusoidal signals with various amplitudes and frequencies, which are shown in Figure 12.

When the 2nd-order compensation was applied to a complex signal, in contrast to the 1st-order compensation, the quadratic polyfit was found to outperform the linear baseline fitting (see Figure 13).

These nonlinearity compensation methods have different results for different nonlinearities strengths. For the 1st-order nonlinearity, a larger initial deviation corresponds to a higher RMS deviation after the compensation. A demonstration of this effect is shown in Figure 14a. For the 2nd-order nonlinearity, the relation between the original distortion and the compensation can be seen in Figure 14b. In both cases, the vibration signal is the one described in Figure 11 with A=3.5×λ/2. The difference of the final compensation results for different original errors can be clearly seen in these figures.

For the data with both nonlinearities, the compensation can be acheived by applying the two algorithms sequentially. The compensation results for applying the orders of the 1st and 2nd nonlinearity compensation methods are shown in Figure 15a. The vibration signal is the one described in Figure 11 with A=3.5×λ/2. The compensated signal of two compensation orders are shown in this figure. The results validate our claim that the 2nd-order nonlinearity compensation should be applied before the 1st-order. The effect of cascaded compensations is also studied and shown in Figure 15b. It is seen that two cycles of compensation with the same method can further improve the results. However, it is also shown that this compensation will not provide an effective improvement anymore after applying it twice.

In practice, there is also a lot of noise in the signal. The impact of noise on the compensation results is shown in Figure 16. The vibration signal is the same as described in Figure 11 with A=3.5×λ/2, and a two-cycle compensation is used in this case. It can be seen that the final compensation results depend on the strength of the noise: a higher noise means a worse compensation. When noise is weak, deviations for different 1st-order errors can be different, but their values will become the same when the noise is higher. The 2nd-order nonlinearities, however, have different phenomena. After a two-cycle compensation, the deviation caused by different 2nd-order nonlinearities is still different when the noise level is high.

We have typical measurement data of some signals retrieved on a silicon-photonics-based homodyne LDV [21] with a sampling rate of 5.5 Msps. The detected vibration is from a silicone block fixed on a stage. During the measurement, the optical table where the silicone is installed is tapped by the operator with his hand. The silicone will shake as a result of the tapping at its resonance frequency (about 800 Hz), which can be detected by the LDV. Since the surface of the silicone is flat and can provide good specular reflection, the sensing light is directly reflected by the silicone surface. The on-chip LDV also has a typical nonlinearity, which can be seen in Figure 17a. We used the Heydemann method and our proposed method to compensate for the errors in the data, the compensation results are also shown in Figure 17a. It can be seen that both Heydemann’s and our methods can remove the clear nonlinear signals by similar RMS values (0.043 ×λ/2 for Heydmann’s correction and 0.044 ×λ/2 for our method). In the spectra, it is seen that the deviation introduced by the nonlinearity is mainly located at around 30 kHz. With both compensation methods, these nonlinear errors can be suppressed.

## 5. Conclusions

This paper addresses the issue of nonlinearities in optical interferometry systems and presents two numerical methods for compensating for these distortions on the demodulated data. The main advantage of the proposed technique is that it does not need to have access to the raw data of the interferometry system, which is usually hidden in the decoder of the interferometry. The proposed methods only need the demodulated data. The first method, called the inverse function method, relies on the periodic nature of the demodulated data and retrieves the strength of the nonlinearity by applying an inverse mapping and analyzing the Fast-Fourier Transform spectrum. The second method, called the piece-wise fitting method, decomposes the signal into sections, fits baseline functions to each section, calculates the deviations, and performs averaging and fitting to estimate the nonlinear parameters. Since the piece-wise fitting method is more versatile than the inverse function method, the analysis is mainly focused on the piece-wise fitting method.

We compare the performance of the piece-wise fitting in compensating for 1st-order and 2nd-order nonlinearities. We also discuss the impact of different baseline fitting functions, the order of compensation, and the presence of noise on the compensation results. We show that it is better to choose the linear baseline fitting for the 2nd-order nonlinearity compensation while choosing the quadratic baseline fitting for the 1st-order compensation. To cascade the compensation for a more general signal, the 2nd-order compensation should be made before the 1st-order. Experimental data from a silicon-photonics-based homodyne LDV validate the effectiveness of the proposed methods in removing nonlinear signals and reducing RMS errors.

The proposed piece-wise fitting method, while innovative, still relies on complex algorithms for the fitting process. To enhance this approach, it is important to develop more efficient algorithms in future work.

## Figures and Tables

**Figure 1 sensors-23-07942-f001:**
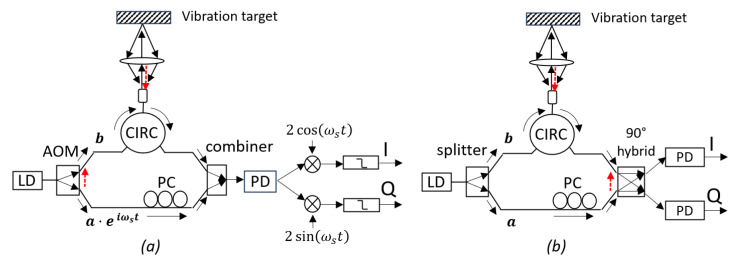
Basic structures of heterodyne and homodyne interferometers for vibration detection and the corresponding sources of nonlinearity: (**a**) heterodyne, and (**b**) homodyne. Solid lines with arrows indicate the main optical beams in the interferometry while the dashed lines indicate the major cross talks that can generate nonlinearities. Here “I” and “Q” denote the in-phase and quadrature signals of the output, respectively, ”CIRC” denotes the circulator, “LD” denotes the laser diode, “PD” denotes the photodiode, and “PC” denotes the polarization controller.

**Figure 2 sensors-23-07942-f002:**
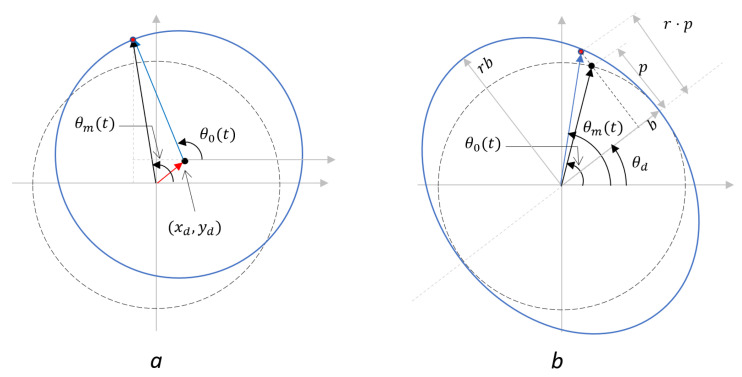
IQ Lissajous curve of (**a**) the 1st-order and (**b**) the 2nd-order nonlinear distortions.

**Figure 3 sensors-23-07942-f003:**
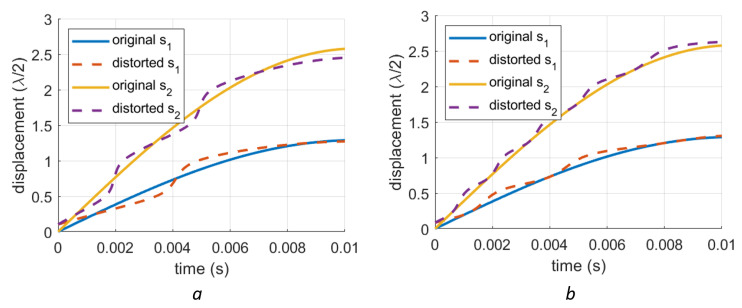
Exaggerated nonlinear phase errors for different vibration amplitudes. Vibration $s_{1}$ is for a vibration with an amplitude of 1 μm while $s_{2}$ is for an amplitude of 2 μm: (**a**) Distorted signals with a 1st-order distortion. The drift of the circle origin is 75% of the radius of the IQ circle. (**b**) Distorted signals with a 2nd-order distortion. In this plot, the eccentricity of the ellipse is 0.9657.

**Figure 4 sensors-23-07942-f004:**
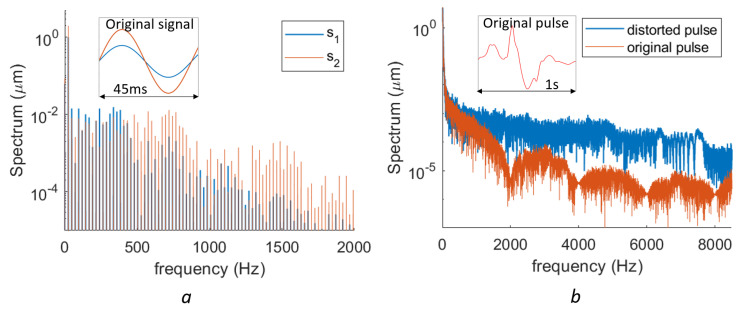
(**a**) Spectra difference of two different signals with the same 2nd-order nonlinearity error but with different vibration amplitudes ($s_{1}\to$ 1 μm, $s_{2}\to$ 2 μm). (**b**) The spectra of a carotid pulse-generated skin displacement and the same signal but with an artificially 2nd-order nonlinear error (e=0.9657).

**Figure 5 sensors-23-07942-f005:**
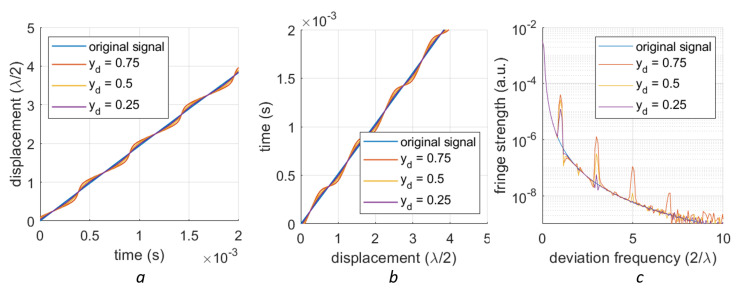
Inverse function of the 1st-order nonlinearity: (**a**) A section of the data that is monotonic (**b**) The inverse function of this section (**c**) The spectrum of the inversely mapped signals, featuring peaks at multiples of 2/λ.

**Figure 6 sensors-23-07942-f006:**
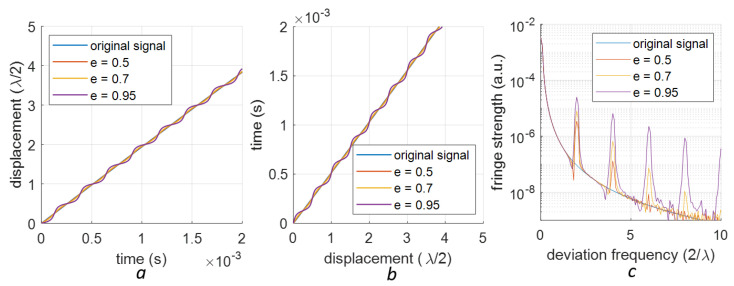
Inverse function of the 2nd-order nonlinearity: (**a**) A section of the data that is monotonic (**b**) The inverse function of this section (**c**) The spectrum of the inversely mapped signals, featuring peaks at multiples of 4/λ.

**Figure 7 sensors-23-07942-f007:**
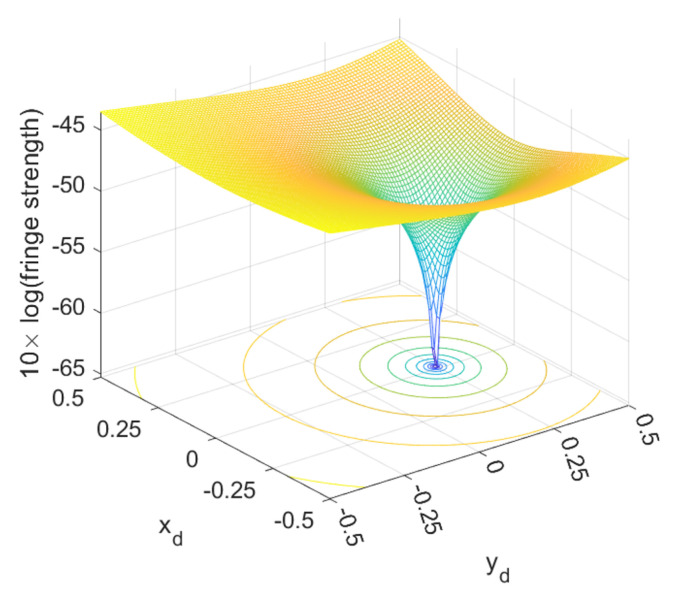
The 2/λ peak values as a result of the scanning of $x_{d}$ and $y_{d}$. The original signal was set with the 1st-order nonlinear error by introducing a $y_{d}$ of 0.25, which is identified through scanning.

**Figure 8 sensors-23-07942-f008:**
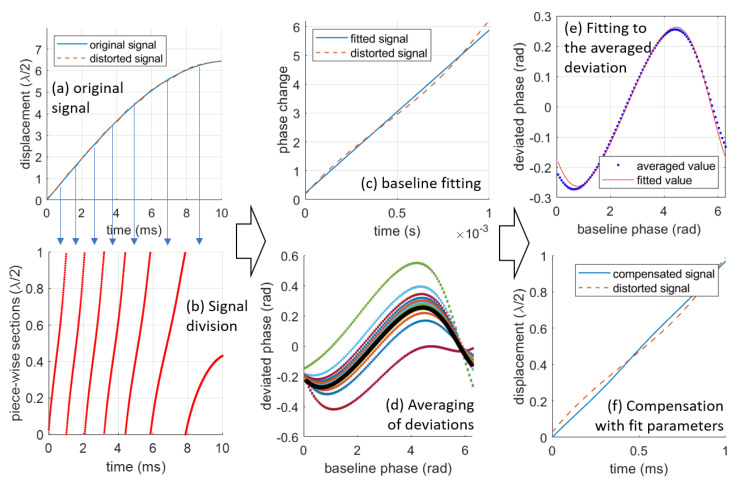
The complete procedure of the 1st-order nonlinear deviation with $x_{d}=-0.2$ and $y_{d}=0.14$ for a signal with an amplitude of 5μm. The compensated parameters are $x_{fd}=-0.223$ and $y_{fd}=0.132$, respectively. The curves in figure (**d**) are the deviations of all complete sections.

**Figure 9 sensors-23-07942-f009:**
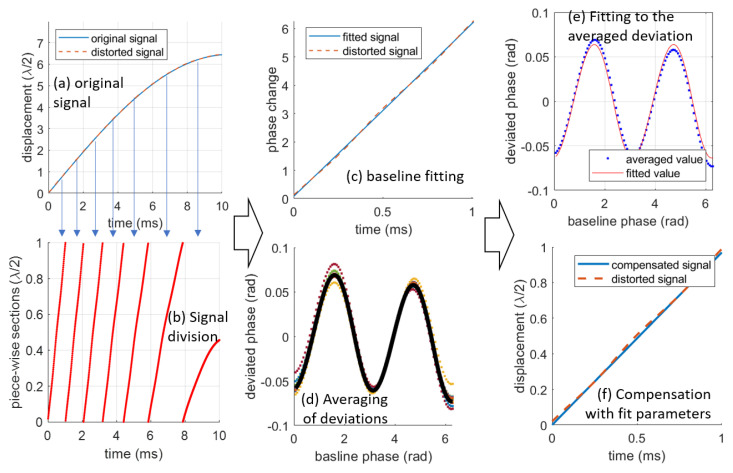
The complete procedure of the 2nd-order nonlinearity compensation for a nonlinear signal with r=1.134 and $\theta =45^{\circ }$ for a signal with an amplitude of 5 μm. The compensated parameters: $r_{f}=1.134$ and $\theta_{fd}=45.0^{\circ }$. The curves in figure (**d**) are the deviations of all complete sections.

**Figure 10 sensors-23-07942-f010:**
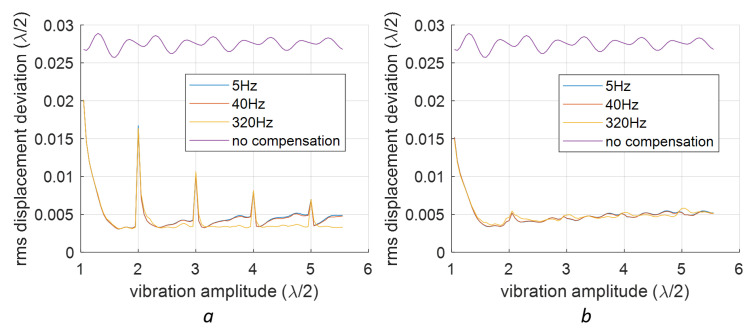
Compensation for the 1st-order nonlinearity at different sinusoidal vibration frequencies and different amplitudes: (**a**) is for linear polyfit and (**b**) is for the quadratic polyfit.

**Figure 11 sensors-23-07942-f011:**
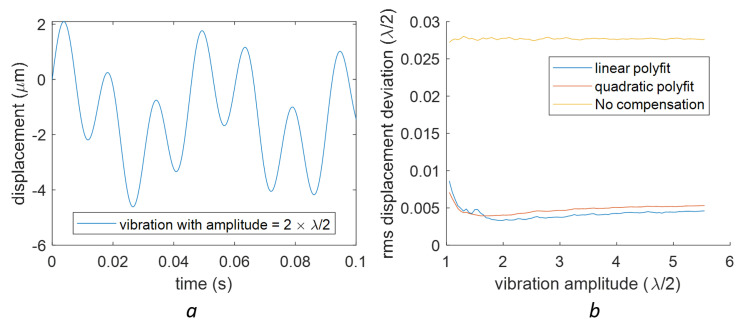
(**a**) A complex signal with two frequencies. The signal is d(t)=A·sin(2πt×20Hz+0.321π)+1.2A·sin(2πt×66.4Hz). In this plot, A=2×λ/2. (**b**) The RMS deviation of the 1st-order error after the nonlinear compensation with a linear baseline fitting and quadratic baseline fitting for different values of *A* (vibration amplitude).

**Figure 12 sensors-23-07942-f012:**
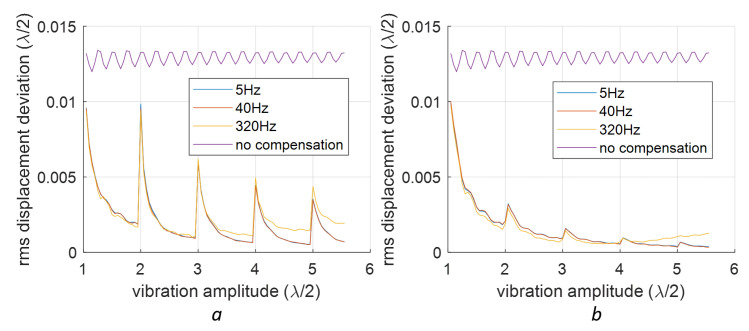
Compensation for the 2nd-order nonlinearity in sinusoidal vibrations at different frequencies and amplitudes. Two baseline fittings are tested: (**a**) linear polyfit; (**b**) quadratic polyfit.

**Figure 13 sensors-23-07942-f013:**
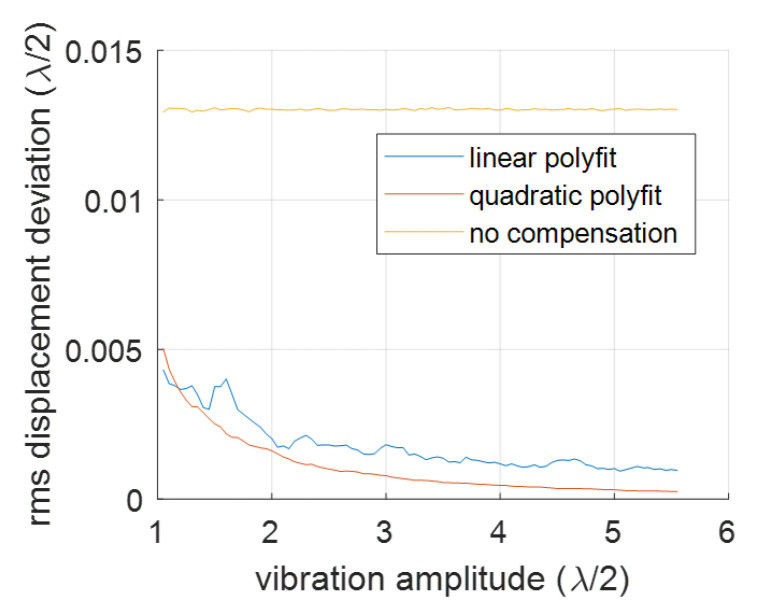
The RMS deviation of the 2nd-order error after the nonlinear compensation on a complex signal with two different baseline fittings (linear fitting and quadratic fitting). The signal is the same as in Figure 11.

**Figure 14 sensors-23-07942-f014:**
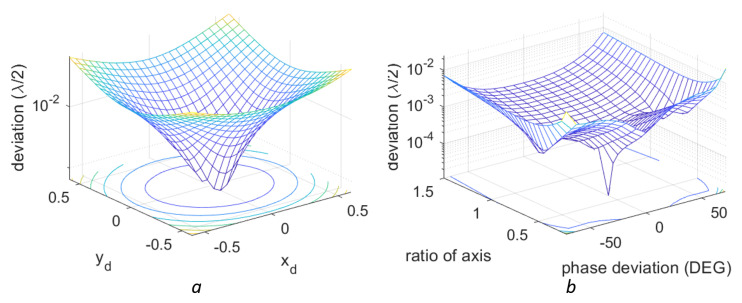
RMS deviation after compensation of the 1st- and 2nd-order nonlinearities with different original distortions. The original signal is the complex vibration shown in Figure 11. (**a**) signals with only the 1st-order nonlinearities (**b**) signals with only the 2nd-order nonlinearities. The colors represent different deviation values.

**Figure 15 sensors-23-07942-f015:**
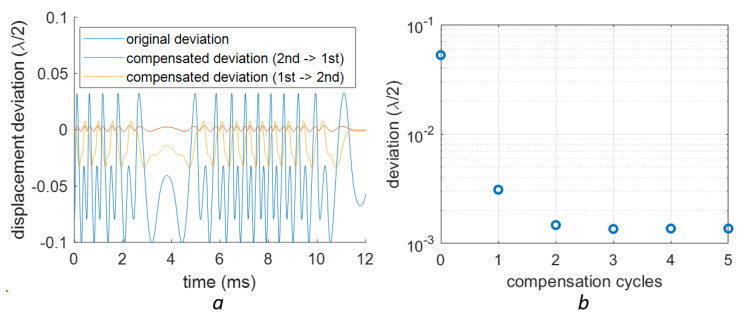
The deviation after compensation on a complex signal shown in Figure 13. (**a**) the deviation in the compensated signal in the time domain using two different orders of compensation for the 1st- and 2nd-order nonlinearities (**b**) the RMS deviation after several cycles of compensation, while each cycle includes one 2nd-order and one 1st-order compensation.

**Figure 16 sensors-23-07942-f016:**
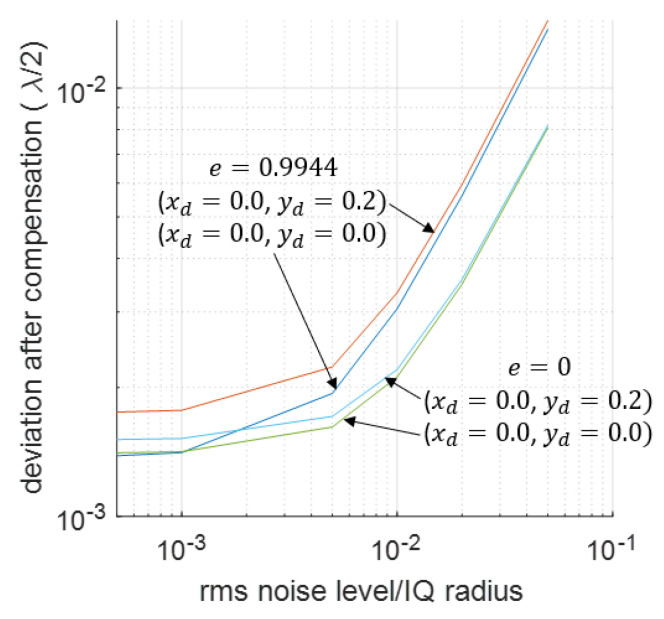
RMS value of the deviation after compensation when noise exists. The effectiveness of the compensation for different noise levels and different original deviation strengths are shown.

**Figure 17 sensors-23-07942-f017:**
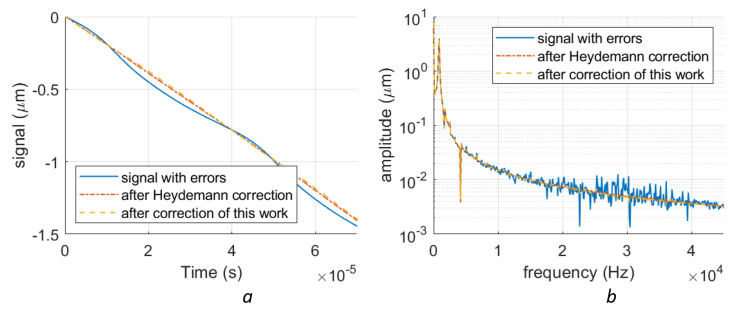
The compensation of a real vibration with nonlinear errors measured with a PIC-based LDV. (**a**) shows the original distorted signal in the time domain, the corresponding signal after Heydemann correction, and the signal after the piece-wise fitting correction. (**b**) The original distorted signal, the signal after Heydemann correction, and the signal after the piece-wise fitting correction in the frequency domain.

## Data Availability

Data underlying the results presented in this paper are not publicly available at this time but may be obtained from the authors upon reasonable request.

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
