# Peer review of "A Compensation Method for Nonlinearity Errors in Optical Interferometry"

_sensors, 2023, doi:10.3390/s23187942_

Round 1

Reviewer 1 Report

Minor typos: l 159 proposed -> Proposed, l 216 can only present -> can only be present, caption fig 16 in the domain -> in the time domain, fig 7e) basline -> baseline

Fig 7b) is the plotted curve the measured signal, then it should be red

Title: is "post-compensation" a good description?

Author Response

Thanks for the reviewer for the suggestions.

The typos are corrected. The typo and the plot color in Fig 7 (now Fig.8) are corrected as suggested. Similar plots are present in Fig 8 (now Fig 9.), and they are also corrected.

The “post-processing” is indeed confusing. The title is changed to: A compensation method for nonlinearity errors in optical interferometry.

Reviewer 2 Report

Please refer to the PDF file.

I think that some sentences in this manuscript are complex and not very fluent.

Author Response

Thanks for the review and comments of the paper. Our replies to the comments are listed below:

  1. line 10. You’d better introduce the specific results of simulation and experiment.

It is indeed good to introduce the specific results for simulation and experiment in the abstract. We will add the following part in the abstract:

Simulation shows that the proposed method can effectively reduce the deviation caused by the nonlinearities without using the raw data. Experimental results from a silicon-photonics-based homodyne laser Doppler vibrometry prove that this method has a similar performance as the conventional Heydemann correction method.

  1. line 11. Mention the homodyne laser interferometry in the abstract.

Both homodyne and heterodyne are now mentioned in the abstract.

  1. Figure 3. Please introduce in detail the changes in nonlinear phase errors for different vibration amplitudes and different orders.

The following parts have been added to the manuscript:

The distortions show a period of \lambda/2 in the displacement domain. Thanks to this phenomenon, the distortion of a signal with a larger amplitude has a higher frequency in the time domain, which can be seen in figure3(a).

The deviations have a displacement periodicity of $\lambda/4$ and their shapes are different from those of the 1st order deviations.

  1. line 186. Add a 2nd-order error diagram similar to Figure 5.

A similar figure (new figure 6) is added to plot the error diagram of the 2nd-order errors.

Similarly, signals with different 2nd-order deviations (eccentricities e = 0.5, 0.7 and 0.95), their inverse functions and the spectra of their inverse functions are shown in figure~\ref{fig4_2}(a), figure~\ref{fig4_2}(b), and figure~\ref{fig4_2}(c), respectively. The periodic features are located at multiples of 4/\lambda. Simulation shows that a higher eccentricity corresponds to a stronger peak at 4/\lambda.

  1. line 244. The amplitude of 6 um does not match to the in Figure 7. The fitted function in line 251 also does not match.

The amplitude of 6 um in the text is indeed a wrong number. It should be 5 um. Similarly, we also forgot to use a correct number in the text. The correct numbers in the text are corrected. We change this in the paper.

  1. line 319. Please specifically introduce the phenomenon of 2nd-order nonlinearities.

I have added the following part:

It can be seen that, after a two-cycle compensation, the deviation caused by different 2nd-order nonlinearities are still different when the noise level is high, that is different from that caused by the 1st order deviations.

Figure 16 (was figure 15) is changed because the old one uses a different vibration signal. Here we want to use the same one as the previous signals. The detailed information of the vibration signal is also describe in this paragraph.

  1. line 352. You can explain the advantages and areas for improvement of the proposed method through comparison.

We added the advantages of these methods in the last paragraph:

The main advantage of the proposed techniques is that it does not need to have access to the raw data of the interferometry system, which is usually hidden in the decoder of the interferometry. The proposed methods only need the demodulated data.

The proposed piece-wise fitting method, while innovative, still relies on complex algorithms for the fitting process.  To enhance this approach, it's important to develop more efficient algorithms

Reviewer 3 Report

The authors present their work on post-compensation method for nonlinearity errors in optical interferometry. Theoretical analyses of nonlinearity errors from interferometer and detection schemes have been carried out in detail. The results in the manuscript clearly show a good work have been done. The paper is technically sound and is well written and organized.

Since the vibration is specified with an amplitude from 1 µm to 10 µm in the manuscript the laser wavelength needs to be given in the analysis to show a clear image of the system.

The manuscript is acceptable in its present form.

Author Response

We would like to thank the reviewer for the review. The wavelength used in simulation and experiment is 1550 nm. We have added this information in the second paragraph of section 3.1 in the new draft.

Reviewer 4 Report

The manuscript is very interesting and reveals the new and useful approach to the correction of the systematic errors in the measuring interferometers. These results are definitely worth publishing in the high-level journal. We have not revealed any serious mistakes, drawbacks or flaws in the presented theory. However, to our opinion the manuscript has yet to be seriously revised. The reviewed version of the manuscript looks like the text written by theoreticians for theoreticians or, at least, for academic scientists. However, to our opinion the real audience of the future paper will be the practical scientists and optical engineers. For them this text is like a “skeleton” without “meat”. The interesting and original theory has to be supplied by practical comments and examples. Let us outline some proposals

1.        Figure 1 is too formal and is overloaded by formulae. To our opinion it has to be accompanied or replaced by another figure with the normal ray tracing and description of elements.

2.       We need some practical examples, assisting the reader in his understanding of the formal methods, developed by the manuscript authors. For instance, it would be great to depict and analyze a couple of more or less realistic interferometers – e.g. two cases, corresponding to the theoretical problems, depicted in the Fig.2.

3.       Similarly, it would be reasonable to outline the practical situations which can lead to the arrival of -1st, 0th, 2nd etc. harmonics.

4.       The procedure of compensation of nonlinearities has also to be outlined in the form of some algorithm, i.e. the sequence of mathematical actions.

We do not see any serious flaws in the current English of the manuscript.

Author Response

Thank you very much for the review. As suggested by the reviewer, we have tried to add more examples to the manuscript. 

  1. Figure 1 is too formal and is overloaded by formulae. To our opinion it has to be accompanied or replaced by another figure with the normal ray tracing and description of elements.

We were trying to describe the problem in more general interferometry systems, but it may be not so easy to understand for some readers. Therefore, we replace figure 1 with layout of some practical optical interferometers using fiber systems. Free-space interferometry usually use polarization beam splitters, which will bring unnecessary details to the paper. Therefore, we will not use examples based on free-space optics.

The descriptions of the systems are also adapted accordingly.

  1. We need some practical examples, assisting the reader in his understanding of the formal methods, developed by the manuscript authors. For instance, it would be great to depict and analyze a couple of more or less realistic interferometers – e.g. two cases, corresponding to the theoretical problems, depicted in the Fig.2.

The practical examples of nonlinear errors will be added based on the fiber-based interferometry. These two cases are two typical problems in optical interferometers. The following text is added to the manuscript

For example, in a homodyne system, the 1st order nonlinearity usually happens when the PD responsivities are not equal, while the 2nd-order nonlinearities happen when the phase relation of  the 90$^\circ$ hybrid is not perfect.

  1. Similarly, it would be reasonable to outline the practical situations which can lead to the arrival of -1st, 0th, 2nd harmonics.

The practical situations of the harmonics will be explained in the paragraph in line 88. The added text is:

The unwanted harmonics generated by the OFS is the first issue. The purpose of using an OFS is to create a constant frequency shift.  AOM is  the most typical OFS. It can generate several harmonics at the same time. The unwanted harmonics are usually removed by only sending the requested frequency shift to the output port of the AOM. However, small amount of other harmonics may still find their ways to the output port, through scattering or reflecting.  One can also generate a frequency shift with other phase modulators (e.g., PN junction based phase modulator in silicon-based photonic integrated circuit~\cite{Xia-2020}) using single-sideband modulation techniques, but these methods usually suffer more from other harmonic orders. Most of these residue high-order harmonics can be removed by applying extra filters.

  1. The procedure of compensation of nonlinearities has also to be outlined in the form of some algorithm, i.e. the sequence of mathematical actions.

The mathematical actions of the compensation procedure of the piece-wise-fitting method was not written, because some similar formulas were shown in equation 17 and 18. But it is better to show the compensation algorithm clearly. Therefore, we add the equation used in the compensation, which can be found in the subsection of the compensation procedure.

Round 2

Reviewer 4 Report

The manuscript can be now accepted.